# Pre-training Multi-task Contrastive Learning Models for Scientific Literature Understanding

**Yu Zhang♣†\*, Hao Cheng♠\*, Zhihong Shen♡, Xiaodong Liu♠, Ye-Yi Wang♡, Jianfeng Gao♠**
♣ University of Illinois at Urbana-Champaign ♠ Microsoft Research
♡ Microsoft Search, Assistant and Intelligence
yuz9@illinois.edu   {chehao,zhihosh,xiaodl,yeyiwang,jfgao}@microsoft.com

## Abstract

Scientific literature understanding tasks have gained significant attention due to their potential to accelerate scientific discovery. Pre-trained language models (LMs) have shown effectiveness in these tasks, especially when tuned via contrastive learning. However, jointly utilizing pre-training data across multiple heterogeneous tasks (*e.g.,* extreme multi-label paper classification, citation prediction, and literature search) remains largely unexplored. To bridge this gap, we propose a multi-task contrastive learning framework, SciMult, with a focus on facilitating common knowledge sharing across different scientific literature understanding tasks while preventing task-specific skills from interfering with each other. To be specific, we explore two techniques – task-aware specialization and instruction tuning. The former adopts a Mixture-of-Experts Transformer architecture with task-aware sub-layers; the latter prepends task-specific instructions to the input text so as to produce task-aware outputs. Extensive experiments on a comprehensive collection of benchmark datasets verify the effectiveness of our task-aware specialization strategy, where we outperform state-of-the-art scientific pre-trained LMs. Code, datasets, and pre-trained models can be found at https://scimult.github.io/.

## 1 Introduction

Scientific literature understanding tasks, such as paper classification (Zhang et al., 2023b), citation prediction (Bhagavatula et al., 2018), scientific literature search (Voorhees et al., 2021), and recommendation (Kanakia et al., 2019), have received increasing attention because they can be broadly applied to academic service platforms (Tang et al., 2008; Sinha et al., 2015; Ammar et al., 2018), and more importantly, uncover knowledge structures to accelerate scientific discovery (Naik et al., 2022; Chandak et al., 2023). Recent studies have demonstrated the effectiveness of pre-trained language

models (LMs) (Beltagy et al., 2019; Gu et al., 2021; Liu et al., 2022) in these tasks as they generate high-quality scientific text representations, especially when the LMs are further tuned via contrastive learning. For example, MICoL (Zhang et al., 2022) proposes a metadata-induced contrastive learning that can perform extreme multi-label paper classification with more than 10,000 classes; SPECTER (Cohan et al., 2020) and SciNCL (Ostendorff et al., 2022) leverage citation information to create training pairs and achieve remarkable performance in predicting various types of links between papers.

Nevertheless, jointly using data across different scientific literature understanding tasks for LM pre-training remains largely unexplored. Intuitively, there are some common knowledge and skills that can be shared across related tasks. For example, accurately identifying fine-grained topic classes of a paper not only helps classification but also provides hints to link prediction and search. Therefore, a multi-task learning framework is expected to be beneficial with improved parameter efficiency. However, if all parameters of the backbone LM are shared across tasks (Liu et al., 2019), the model is observed to suffer from the undesirable *task interference* (Ma et al., 2023), that is, the model sacrifices the performance on some tasks to boost the others when jointly trained to a certain extent. This is because specialized skills are still required in different tasks competing for the limited shared parameter space. For instance, the encoder for extreme multi-label paper classification should focus more on fine-grained fields-of-study entities in each paper, while the encoder for citation prediction should put more effort into understanding citation intents. Mixing these two skills may result in a negative transfer across the two tasks.

In this paper, to mitigate task interference in multi-task scientific literature understanding, we propose to consider two techniques: *task-aware specialization* and *instruction tuning*. Task-aware specialization, inspired by the Mixture-of-Experts (MoE) Transformer architecture (Fedus et al., 2022;

---

\* Equal contribution
† Work done during an internship at Microsoft Research.

Du et al., 2022; Zhou et al., 2022; Cheng et al., 2023), modifies the Transformer block in the LM to have multiple parallel sub-layers, each of which is dedicated for one task. When performing different tasks, the input will be routed to different sub-layers based on task types. In this way, the encoder contains both shared parameters and task-specific parameters, making it capable of producing task-aware outputs when tapping into shared knowledge. In contrast, instruction tuning adopts one encoder for all tasks, but it prepends task-specific instructions (Wei et al., 2022; Sanh et al., 2022; Ouyang et al., 2022; Wang et al., 2022; Chung et al., 2022; Asai et al., 2023) to the input text during training so that the encoder can learn to produce task-aware representations. These two ideas are different from the techniques (*e.g.,* adapters (Houlsby et al., 2019) and control codes (Keskar et al., 2019)) explored in previous studies (Singh et al., 2022) for multi-task scientific text representation learning. Indeed, as far as we know, this is a pioneering study that explores the effect of the MoE Transformer architecture and instruction tuning in scientific NLP.

To validate the efficacy of our proposed techniques, we conduct a comprehensive empirical study using datasets from multiple sources (Kanakia et al., 2019; Cohan et al., 2020; Thakur et al., 2021; Zhao et al., 2022; Singh et al., 2022; Zhang et al., 2023b) for evaluating various scientific literature understanding tasks. For each task, models will be tested on not only *in-domain* but also *cross-domain* evaluation datasets. Specifically, for extreme multi-label text classification, models trained on computer science and biomedicine papers will be tested in the geography and psychology fields (Zhang et al., 2023b); for link prediction, models trained on citation signals (Cohan et al., 2020) need to be evaluated on patient summary retrieval (Zhao et al., 2022) and paper recommendation (Kanakia et al., 2019); for search, models will be tested on datasets specific to COVID-19 (Voorhees et al., 2021) or related to claim verification (Wadden et al., 2020) which are not seen during pre-training. Experimental results show that SciMult-MHAExpert outperforms competitive scientific pre-trained LMs (Cohan et al., 2020; Ostendorff et al., 2022; Singh et al., 2022) on most datasets and achieves the new state-of-the-art performance on the leaderboard of PMC-Patients (Zhao et al., 2022). Ablation studies further prove that task-aware specialization can effectively mitigate task interference, while the improvement brought by instruction tuning is not consistent across all tasks.

## 2 Background

We consider three widely studied tasks in scientific literature understanding: classification, link prediction, and search.

**(Extreme Multi-Label) Classification.** Classifying academic papers to their relevant label(s) is a fundamental task in scientific text mining. It can help organize papers according to their fields/themes and benefit downstream applications such as trend analysis of scientific topics (Prabhakaran et al., 2016; Jin et al., 2021). The label space $\mathcal{L}$ can be either coarse-grained (Cohan et al. 2020; *e.g.,* predicting whether a paper belongs to "Computer Science" or "Biology") or fine-grained (Peng et al. 2016; Xun et al. 2019; Ye et al. 2021; Zhang et al. 2021, 2023b; *e.g.,* predicting whether a paper is relevant to "Alphacoronavirus" or "Betacoronavirus" or both). When $\mathcal{L}$ is large and fine-grained (*e.g.,* with more than 10,000 labels), it is natural to assume that each paper $p$ can be relevant to more than one label. This task is called extreme multi-label classification (Liu et al., 2017; Prabhu et al., 2018; Chang et al., 2020), which aims to rank all labels $l \in \mathcal{L}$ according to how likely $p$ is relevant to $l$.

**Link Prediction.** Link prediction aims to predict if a certain type of link exists between a query paper $p_Q$ and a candidate paper $p_C$. Narrowly, "links" refer to citation links (Bhagavatula et al. 2018; Wright and Augenstein 2021; *i.e.,* $p_Q$ cites $p_C$). Broadly, "links" can be defined as relations that $p_Q$ and $p_C$ are co-viewed frequently by users, co-cited frequently by other papers, and so on (Cohan et al., 2020; Zhao et al., 2022). An accurate link prediction model can benefit tasks like paper recommendation (Kanakia et al., 2019) and help identify the potential use of scientific literature (Yin et al., 2022; Lin et al., 2023).

**Search.** Scientific literature search helps researchers track their interested fields-of-study and prevents them from drowning in the whole literature. Given a search query $q$ and a pool of papers $\mathcal{P}$, the task is to find papers $p \in \mathcal{P}$ that are relevant to $q$. Search also serves as the initial step of more complex scientific text mining tasks, such as claim verification (Wadden et al., 2020) and open-domain question answering (Jin et al., 2019).

## 3 Models

### 3.1 Multi-task Contrastive Learning

One can observe that the three tasks bear a common feature – a "query" $q$ and a pool of "candidates"

$\mathcal{C} = \{c_1, c_2, ..., c_{|\mathcal{C}|}\}$ are given. To be specific, $q$ is the paper $p$ to be classified in classification, the query paper $p_Q$ in link prediction, and the search query $q$ in literature search; $\mathcal{C}$ is the label space $\mathcal{L}$ in classification, the set of candidate papers $p_C$ in link prediction, and the candidate paper pool $\mathcal{P}$ in search. This motivates us to jointly train a multi-task model that is applicable to all tasks.

To implement this idea, our proposed `SciMult` framework is built upon a Bi-Encoder architecture, where two encoders (whose parameters are shared) encode queries and candidates independently. Following Karpukhin et al. (2020), we adopt maximum inner product search (MIPS) in the embedding space to find positive candidates for each query. Formally, the similarity is calculated as follows:

$$\text{sim}(q, c) = \mathbf{E}(q)^\top \mathbf{E}(c), \tag{1}$$

where $\mathbf{E}(\cdot)$ is a text encoder (*e.g.,* a Transformer-based LM) to be learned. We assume the text information of a paper is its title and abstract (*i.e.,* "[CLS] {title} [SEP] {abstract} [SEP]"). Following Zhang et al. (2022), we assume that each label's name and definition are available[1], which constitute the label text information (*i.e.,* "[CLS] {name} [SEP] {definition} [SEP]"). For search queries, we naturally form their text input as "[CLS] {query} [SEP]". The output of $\mathbf{E}(\cdot)$ is the representation of [CLS] after the last layer.

The Bi-Encoder is trained via a contrastive learning objective by pulling relevant $q$ and $c_+$ together while pushing irrelevant $q$ and $c_-$ apart:

$$\min_{\mathbf{E}(\cdot)} -\log \frac{\exp(\text{sim}(q, c_+))}{\exp(\text{sim}(q, c_+)) + \sum_{c_-} \exp(\text{sim}(q, c_-))}. \tag{2}$$

Although it has been shown that model parameter sharing improves the performance of Bi-Encoder retrievers (Xiong et al., 2021), simply sharing all parameters of the backbone LM (Liu et al., 2019) may suffer from task interference and lead to suboptimal performance. This is because the semantics implied by relevant $(q, c)$ pairs for different tasks vary. For example, the encoder for fine-grained paper classification needs to pay more attention to fields-of-study entities, while the encoder for link prediction focuses on understanding

---

[1]This assumption holds for benchmark scientific label spaces such as fields-of-study in the Microsoft Academic Graph (MAG) (Shen et al., 2018) and terms in Medical Subject Headings (MeSH) (Coletti and Bleich, 2001). Meanwhile, if label definitions are not available, our model can take label names as the only input, the performance of which is studied in Appendix E.4.

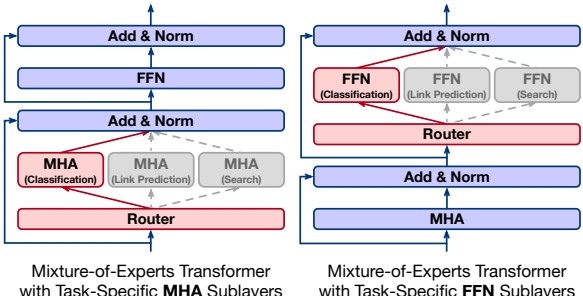

Figure 1: Two different types of Mixture-of-Experts Transformer architecture. They will route the input to different MHA and FFN sub-layers, respectively, when considering different tasks.

citation intents. To tackle this issue, we consider two different strategies, *task-aware specialization* and *instruction tuning* to learn task-specific representations of scientific text.

### 3.2 Task-Aware Specialization

Inspired by recent Mixture-of-Experts (MoE) models (Fedus et al., 2022; Du et al., 2022; Zhou et al., 2022; Cheng et al., 2023; Ma et al., 2023), we propose to adopt task-specific Transformer blocks in the LM architecture. Specifically, a typical Transformer block (Vaswani et al., 2017) contains a multi-head attention (MHA) sub-layer and a feed-forward network (FFN) sub-layer. Fedus et al. (2022) propose a Mixture-of-Experts Transformer block with multiple FFN sub-layers stacked upon a shared MHA sub-layer. As shown in Figure 1 (right), we adopt this architecture and let each FFN sub-layer correspond to one particular task $t$ ($t \in \{$classification, link prediction, search$\}$). For example, if the encoder $\mathbf{E}(\cdot)$ is trained/tested on a classification task, the input will be routed to the classification FFN. Different from Fedus et al. (2022), Ma et al. (2023) propose to specialize the MHA sub-layer and observe better performance in open-domain question answering. We test this architecture as well, where, as shown in Figure 1 (left), a shared FFN sub-layer is stacked upon task-specific MHA sub-layers. We again use the same task-dependent routing for this variant.

In `SciMult`, following Du et al. (2022), we stack typical Transformer blocks and task-specific Transformer blocks alternately, *i.e.,* for a base-size LM with 12 Transformer blocks, there will be 6 typical Transformer blocks and 6 task-specific Transformer blocks interleaved with each other. In this way, the encoder will have both parameters $\theta_t$ characterizing specific skills for task $t$ and parameters $\theta$ characterizing the common knowledge shared across all tasks. Thus, we denote the encoder as $\mathbf{E}_{\{\theta, \theta_t\}}(\cdot)$,

and the task-aware similarity between the query $q$ and the candidate $c$ will be modified as

$$\text{sim}_t(q, c) = \mathbf{E}_{\{\theta, \theta_t\}}(q)^\top \mathbf{E}_{\{\theta, \theta_t\}}(c). \quad (3)$$

Note that the query encoder and the candidate encoder for the same task still share their parameters.

## 3.3 Instruction Tuning

Training LMs with task-specific instructions (Wei et al., 2022; Sanh et al., 2022; Ouyang et al., 2022; Wang et al., 2022; Chung et al., 2022; Asai et al., 2023) has been extensively studied with remarkable progress achieved. However, the effect of instruction tuning on scientific literature understanding tasks has remained elusive. Moreover, the major focus of previous studies is on effective zero-shot or few-shot model transfer to new tasks rather than mitigating task interference. As a result, a head-to-head comparison between instruction tuning and MoE in multi-task LM pre-training is missing, so we aim to bridge the gap in this paper.

Different from task-aware specialization which trains $\mathbf{E}_{\{\theta, \theta_t\}}(\cdot)$ for each task $t$, instruction tuning keeps one encoder $\mathbf{E}_\theta(\cdot)$ with all its parameters $\theta$ shared across different tasks. Each task $t$ is instead characterized by a natural language instruction $x_t$. The instructions we use for the three tasks are shown in Table 1. We can prepend the representations of $x_t$ to the query and candidate texts to get their task-aware embeddings. To be specific, suppose $x_t$ contains $K$ tokens $\{x_{t,k}\}_{k=1}^K$. We first use an instruction encoder $\mathbf{E}_\phi(\cdot)$ to encode $x_t$ and get its token representations $\{\boldsymbol{x}_{t,k}^{(n)}\}_{k=1}^K$ after each layer $n \in \{0, 1, ..., N\}$. Then, we use the query/candidate encoder $\mathbf{E}_\theta(\cdot)$ to encode $q = q_1 q_2 ... q_A$ and $c = c_1 c_2 ... c_B$. At layer $n \in \{1, 2, ..., N\}$, the output representations of $q$ and $c$ will take the instruction token representations corresponding to that layer as context. Formally,

$$
\begin{aligned}
\boldsymbol{q}_1^{(n)}, ..., \boldsymbol{q}_A^{(n)} &= \text{Transformer}(\boldsymbol{x}_{t,1}^{(n-1)}, ..., \boldsymbol{x}_{t,K}^{(n-1)}, \\
& \qquad \boldsymbol{q}_1^{(n-1)}, ..., \boldsymbol{q}_A^{(n-1)}), \\
\boldsymbol{c}_1^{(n)}, ..., \boldsymbol{c}_B^{(n)} &= \text{Transformer}(\boldsymbol{x}_{t,1}^{(n-1)}, ..., \boldsymbol{x}_{t,K}^{(n-1)}, \\
& \qquad \boldsymbol{c}_1^{(n-1)}, ..., \boldsymbol{c}_B^{(n-1)}).
\end{aligned} \quad (4)
$$

The task-aware similarity between $q$ and $c$ will then be modified as

$$\text{sim}_t(q, c) = \boldsymbol{q}_{\texttt{[CLS]}}^{(N)}{}^\top \boldsymbol{c}_{\texttt{[CLS]}}^{(N)}. \quad (5)$$

There are two ways to train the model. First, we can update the entire architecture, including both the instruction encoder $\mathbf{E}_\phi(\cdot)$ and the

| Task | Instruction |
|------|-------------|
| Classification | Tag a scientific paper with relevant scientific topic classes. |
| Link Prediction | Find a pair of scientific papers that one paper cites the other. |
| Search | Retrieve a scientific paper that is relevant to the query. |

Table 1: Instructions used for the three tasks.

query/candidate encoder $\mathbf{E}_\theta(\cdot)$, and let them share parameters (*i.e.,* $\phi = \theta$). Second, we can keep the query/candidate encoder frozen and optimize the instruction encoder only, which bears similarities with prefix-tuning (Li and Liang, 2021) by treating instructions as prefixes. We will evaluate both approaches in our experiments.

## 3.4 Negative Sampling

Previous studies on contrastive learning with scientific text (Cohan et al., 2020; Ostendorff et al., 2022) have emphasized the importance of hard negatives. Specific to citation prediction, Cohan et al. (2020) propose a way to derive hard negatives: Given a positive query-candidate pair $(p_Q, p_{C+})$ where $p_Q$ cites $p_{C+}$, if there exists a paper $p_{C-}$ such that (1) $p_{C+}$ cites $p_{C-}$ but (2) $p_Q$ does not cite $p_{C-}$, then $p_{C-}$ is a hard negative.

We generalize this idea to the other tasks. For extreme multi-label classification, given a positive paper-label pair $(p, l)$, we consider a paper $p'$ cited by $p$, sharing a common author with $p$, or published in the same venue as $p$, in which case $p'$ should be semantically close to $p$. If $p'$ has a label $l'$ that is irrelevant to $p$, then $l'$ is treated as a hard negative for $(p, l)$. For literature search, we use the training data from Singh et al. (2022) which contains a short list of papers returned by an academic search engine for each query $q$. The papers clicked by users will be treated as positives $p_+$, and the others in the list will be viewed as hard negatives $p_-$.

Related studies (Cohan et al., 2020; Ostendorff et al., 2022) show that combining easy negatives (*i.e.,* negatives randomly sampled from the entire candidate pool $\mathcal{C}$) and hard negatives leads to better performance. Thus, during pre-training, for each positive pair $(q, c_+)$, we sample one hard negative and treat all in-batch negatives (Karpukhin et al., 2020) as easy negatives, both of which are combined as $c_-$'s to optimize Equation 2.

## 4 Experiments

### 4.1 Datasets

We adopt a comprehensive collection of benchmark datasets for model evaluation. Each task has its pre-

| Task | Pre-training | In-domain Evaluation | Cross-domain Evaluation |
|---|---|---|---|
| Classification | **MAPLE** (Zhang et al., 2023b) {CS-Journal, Biology-MeSH, Medicine-MeSH} | **MAPLE** (Zhang et al., 2023b) {CS-Conference, Chemistry-MeSH}, **SciDocs** (Cohan et al., 2020) {MAG Fields, MeSH Diseases} | **MAPLE** (Zhang et al., 2023b) {Geography, Psychology} |
| Link Prediction | **Citation Prediction Triplets** (Cohan et al., 2020) | **SciDocs** (Cohan et al., 2020) {Co-view, Co-read, Cite, Co-cite} | **Recommendation** (Kanakia et al., 2019), **PMC-Patients** (Zhao et al., 2022) |
| Search | **SciRepEval-Search** (Singh et al., 2022) | **SciRepEval-Search** (Singh et al., 2022) | **TREC-COVID** (Voorhees et al., 2021), **SciFact** (Wadden et al., 2020), **NFCorpus** (Boteva et al., 2016) |

Table 2: Datasets used for pre-training, in-domain evaluation, and cross-domain evaluation.

training, *in-domain* evaluation, and *cross-domain* evaluation datasets[2], which will be briefly introduced below. Table 2 summarizes our usage of these datasets with more details in Appendix A.

**Classification.** We consider the MAPLE benchmark (Zhang et al., 2023b), which consists of 23 fine-grained multi-label paper classification datasets across 19 scientific fields. Each paper in MAPLE is tagged with its relevant MAG fields-of-study (Shen et al., 2018) and MeSH terms (Coletti and Bleich, 2001). Among the 23 datasets, CS-Journal, Biology-MeSH, and Medicine-MeSH are selected for pre-training; CS-Conference and Chemistry-MeSH are used for in-domain evaluation; Geography and Psychology, whose candidate label spaces are not seen during pre-training, are utilized for cross-domain evaluation. Besides, we use the MAG and MeSH datasets in the SciDocs benchmark (Cohan et al., 2020) as in-domain evaluation datasets for coarse-grained paper classification.

**Link Prediction.** For pre-training, we leverage more than 819K citation prediction triplets released in Cohan et al. (2020), which were used to pre-train SPECTER (Cohan et al., 2020). For in-domain evaluation, we make use of the SciDocs benchmark (Cohan et al., 2020), which evaluates the prediction of four link types: Co-view, Co-read, Cite, and Co-cite. For cross-domain evaluation, we use (1) the PMC-Patients dataset (Zhao et al., 2022) where each query is a patient summary and the task is to find its linked research articles and patient summaries, and (2) the Recommendation dataset (Kanakia et al., 2019) collected via an online survey, where the participants are authors of query papers, and they need to judge the relevance between the query paper and some candidate papers on a scale of 1 to 5.

**Search.** For pre-training, we exploit the Search dataset released in the SciRepEval benchmark

(Singh et al., 2022) with 528,497 queries. It also has a hold-out testing set with 2,637 queries, which we employ for in-domain evaluation. For cross-domain evaluation, we adopt TREC-COVID (Voorhees et al., 2021), SciFact (Wadden et al., 2020), and NFCorpus (Boteva et al., 2016), all of which are from the popular BEIR benchmark (Thakur et al., 2021). Note that TREC-COVID has two different versions in SciRepEval and BEIR. The BEIR version has more simplified queries and a larger pool of candidate papers. We will report model performance on both versions.

## 4.2 Compared Methods

We consider the following LM baselines: SciBERT (Beltagy et al., 2019), SentBERT (Reimers and Gurevych, 2019), SPECTER (Cohan et al., 2020), PubMedBERT (Gu et al., 2021), LinkBERT (Yasunaga et al., 2022), BioLinkBERT (Yasunaga et al., 2022), OAG-BERT (Liu et al., 2022), SciNCL (Ostendorff et al., 2022), and SPECTER 2.0 (Singh et al., 2022). Details about the baselines can be found in Appendix B.

Besides LM baselines, we also report the performance of some task-specific classical methods, such as Citeomatic (Bhagavatula et al., 2018) for citation prediction, Kanakia et al. (2019) for recommendation, and BM25 (Robertson and Walker, 1994) for search.

For our `SciMult` model, we pre-train five model variants: `SciMult-Vanilla`, `SciMult-MHAExpert`, `SciMult-FFNExpert`, `SciMult-Prefix`, and `SciMult-Instruction`. All of them are initialized from PubMedBERT.[3] Among them, `SciMult-MHAExpert` and `SciMult-FFNExpert` adopt the task-aware specialization strategy presented in subsection 3.2 with task-specific MHA and FFN sub-layers, respectively; `SciMult-Prefix` and `SciMult-Instruction` leverage instruction tuning presented in subsection 3.3, the former of which tunes the instruction encoder only while the latter

---

[2]In-domain evaluation datasets share the same data source and properties (*e.g.,* the label space, the link type) with pre-training data; cross-domain evaluation datasets are otherwise and evaluate model generalizability.

[3]https://huggingface.co/microsoft/BiomedNLP-PubMedBERT-base-uncased-abstract

| Fine-grained classification | MAPLE (Zhang et al., 2023b) | | | | | | | | | | | | |
|---|---|---|---|---|---|---|---|---|---|---|---|---|---|
| | CS-Conference | | | Chemistry-MeSH | | | Geography | | | Psychology | | | |
| | R@20 | R@50 | R@100 | R@20 | R@50 | R@100 | R@20 | R@50 | R@100 | R@20 | R@50 | R@100 | Average |
| SciBERT (Beltagy et al., 2019) | 42.01 | 42.84 | 43.87 | 30.53 | 31.46 | 32.15 | 52.04 | 54.53 | 58.11 | 43.07 | 44.02 | 45.22 | 43.32 |
| SentBERT (Reimers and Gurevych, 2019) | 42.79 | 44.34 | 45.96 | 30.75 | 31.73 | 32.44 | 53.54 | 57.23 | 61.11 | 43.33 | 44.60 | 46.37 | 44.52 |
| SPECTER (Cohan et al., 2020) | 47.38 | 53.18 | 58.43 | 34.26 | 39.35 | 43.41 | 59.12 | 65.33 | 70.75 | 47.07 | 51.30 | 56.17 | 52.15 |
| PubMedBERT (Gu et al., 2021) | 41.93 | 42.56 | 43.24 | 30.46 | 31.46 | 31.83 | 52.19 | 54.82 | 56.88 | 43.93 | 46.28 | 49.27 | 43.74 |
| LinkBERT (Yasunaga et al., 2022) | 42.15 | 43.16 | 44.22 | 30.52 | 31.56 | 32.37 | 50.58 | 50.94 | 51.63 | 42.62 | 42.90 | 43.23 | 42.16 |
| BioLinkBERT (Yasunaga et al., 2022) | 42.00 | 42.81 | 43.57 | 30.37 | 31.15 | 31.48 | 50.36 | 50.54 | 50.86 | 42.39 | 42.55 | 42.79 | 41.74 |
| OAG-BERT (Liu et al., 2022) | 42.59 | 43.79 | 44.93 | 30.58 | 31.97 | 32.62 | 51.44 | 52.25 | 53.16 | 42.63 | 42.95 | 43.30 | 42.68 |
| SciNCL (Ostendorff et al., 2022) | 47.92 | 53.57 | 58.29 | 34.99 | 40.50 | 44.64 | 59.00 | 65.49 | 71.41 | 48.74 | 54.21 | 59.84 | 53.22 |
| SPECTER 2.0 (Singh et al., 2022) | 48.63 | 55.09 | 60.68 | 36.17 | 43.06 | 48.26 | 62.87 | 70.30 | 76.37 | 50.60 | 58.27 | 65.66 | 56.33 |
| SciMult-Vanilla | 53.40 | 64.70 | 74.09 | **39.78** | **51.31** | **59.75** | 62.08 | 70.65 | 77.79 | 50.42 | 56.58 | 63.17 | 60.31 |
| SciMult-MHAExpert | **54.02** | **65.49** | **75.07** | 39.41 | 50.92 | 59.59 | **65.94** | **75.01** | **81.93** | **51.77** | **59.55** | **67.86** | **62.21** |
| SciMult-FFNExpert | 53.73 | 63.79 | 72.46 | 38.01 | 48.76 | 57.43 | 61.90 | 70.69 | 78.81 | 50.09 | 56.94 | 64.28 | 59.74 |
| SciMult-Prefix | 53.68 | 63.62 | 72.07 | 37.97 | 48.95 | 57.56 | 62.86 | 71.65 | 79.71 | 50.10 | 57.25 | 64.53 | 60.00 |
| SciMult-Instruction | 53.78 | 63.99 | 72.72 | 38.81 | 50.12 | 58.96 | 63.26 | 71.74 | 79.52 | 50.86 | 58.47 | 66.46 | 60.72 |

Table 3: Fine-grained classification performance on MAPLE (Zhang et al., 2023b). Blue : In-domain evaluation datasets. Red : Cross-domain evaluation datasets. Gray : Better than all baselines. **Bold**: The best score.

tunes the whole architecture; SciMult-Vanilla uses neither of the two strategies. All variants utilize hard negatives introduced in subsection 3.4 during contrastive learning. Hyperparameter configurations of SciMult can be found in Appendix C.

### 4.3 Fine-grained Classification

Following Zhang et al. (2023b), we consider a simple heuristic when ranking all candidate labels: labels whose name appears in the query paper $p$ should be ranked higher than those not appearing in $p$. In other words, we first rank all labels $l$ according to $sim(p, l)$ and then reorder all labels appearing in $p$ in front of all other labels. Here, we evaluate model performance using Recall@$k$ ($k = 20, 50, 100$), *i.e.,* the proportion of gold labels found in the top-$k$ retrieved results to all gold labels of a paper.

The results are shown in Table 3. We also conduct experiments without using this heuristic, the results of which are shown in Table A3, where all models perform consistently worse. From Table 3, we observe that: (1) SciMult-MHAExpert and SciMult-Instruction outperform all baselines on all four datasets. The other three SciMult variants have significant advantages over all baselines on in-domain evaluation datasets, but their edges on cross-domain datasets are not consistent. (2) Comparing among all the SciMult variants, SciMult-MHAExpert is always the best on cross-domain datasets, indicating its generalizability to unseen label spaces.

### 4.4 Coarse-grained Classification

Following Cohan et al. (2020) and Ostendorff et al. (2022), we directly predict the most likely coarse label of each paper and use Macro-F1 as the evaluation metric. Two different settings are considered

here: (1) The *Bi-Encoder* setting (abbreviated to "BiEnc" in Table 4) follows our practice in fine-grained classification where we calculate $sim_t(p, l)$ given the description of each $l$ (without relying on any training data after LM pre-training). (2) The *Linear Classifier* setting (abbreviated to "Linear" in Table 4) follows the practice in Cohan et al. (2020), which takes the embedding vector $\mathbf{E}(p)$ as the input feature of each paper $p$ and trains a linear SVM for classification. This setting requires labeled training and validation data after LM pre-training but no longer needs label descriptions.

The results of both settings are shown in Table 4. We find that, on average, SciMult variants can always beat all baselines with a more pronounced out-of-box advantage (*i.e.,* the "BiEnc" setting). On the other hand, when further fine-tuned with a linear classifier, most pre-trained models perform very similarly with better performance achieved by the multi-task models (SPECTER 2.0 and our SciMult variants), indicating the advantage of multi-task training for coarse-grained classification.

### 4.5 Link Prediction

In the link prediction task, we also consider two settings: *retrieval* and *reranking*.

For the retrieval setting, given a query paper $p_Q$, the task is to find all candidate papers $p_C$ that are linked to $p_Q$ (via the relation "Cite", "Co-cite", *etc.* ) from the whole dataset. The link prediction performance of compared methods under the retrieval setting is shown in Table 5, where the evaluation metrics are Recall@20, 50, and 100. As expected, SciMult variants significantly outperform all baselines in almost all cases.

As for the cross-domain PMC-Patients dataset (Zhao et al., 2022), besides evaluating the zero-shot retrieval performance (Table 5), we also test

| Coarse-grained Classification | SciDocs (Cohan et al., 2020) | | | | |
| | MAG Fields | | MeSH Diseases | | |
| | Linear | BiEnc | Linear | BiEnc | Average |
|---|---|---|---|---|---|
| BM25 | – | 23.54 | – | 14.84 | – |
| SciBERT | 79.7† | 18.81 | 80.7† | 12.29 | 47.88 |
| SentBERT | 80.5† | 38.76 | 69.1† | 9.07 | 49.36 |
| SPECTER | 82.0† | 55.49 | 86.4† | 57.70 | 70.40 |
| PubMedBERT | 77.44 | 10.67 | 81.46 | 1.14 | 42.68 |
| LinkBERT | 77.51 | 14.00 | 64.75 | 1.75 | 39.50 |
| BioLinkBERT | 76.28 | 7.37 | 85.50 | 1.20 | 42.59 |
| OAG-BERT | 77.64 | 20.87 | 80.84 | 1.91 | 45.32 |
| SciNCL | 81.4† | 60.02 | 88.7† | 58.67 | 72.20 |
| SPECTER 2.0 | **82.66†** | 60.00 | **89.40†** | 66.03 | 74.52 |
| SciMult-Vanilla | 81.31 | 65.63 | 88.92 | 66.29 | 75.54 |
| SciMult-MHAExpert | 81.76 | 67.38 | 89.37 | **66.72** | 76.31 |
| SciMult-FFNExpert | 82.25 | 69.26 | 88.07 | 63.57 | 75.79 |
| SciMult-Prefix | 82.48 | 69.02 | 88.62 | 66.55 | **76.67** |
| SciMult-Instruction | 82.63 | **69.79** | 88.71 | 63.82 | 76.24 |

Table 4: Coarse-grained classification performance (Macro-F1 scores) on SciDocs (Cohan et al., 2020). Scores with † are reported in Cohan et al. (2020), Ostendorff et al. (2022), and Singh et al. (2022).

| Link Prediction (Retrieval) | SciDocs (Cohan et al., 2020) | | | | | | PMC-Patients (2022) | | | |
| | Cite | | | Co-cite | | | Patient-to-Patient | | | |
| | R@20 | R@50 | R@100 | R@20 | R@50 | R@100 | R@20 | R@50 | R@100 | Average |
|---|---|---|---|---|---|---|---|---|---|---|
| BM25 | 17.81 | 24.76 | 30.49 | 24.19 | 31.98 | 38.03 | 42.20 | 48.68 | 53.75 | 34.65 |
| SciBERT | 2.45 | 5.36 | 5.32 | 5.36 | 7.17 | 9.18 | 17.15 | 20.17 | 22.90 | 10.39 |
| SentBERT | 5.45 | 8.29 | 11.13 | 9.19 | 12.65 | 15.93 | 8.20 | 10.46 | 12.52 | 10.42 |
| SPECTER | 21.59 | 31.58 | 40.02 | 31.51 | 43.41 | 52.90 | 27.72 | 34.52 | 40.43 | 35.96 |
| PubMedBERT | 4.42 | 6.46 | 8.43 | 8.42 | 11.21 | 13.77 | 21.42 | 24.48 | 27.18 | 13.98 |
| LinkBERT | 2.38 | 3.47 | 4.60 | 5.02 | 6.33 | 7.69 | 11.77 | 13.66 | 15.45 | 7.82 |
| BioLinkBERT | 3.41 | 5.01 | 6.63 | 6.93 | 9.14 | 11.24 | 21.28 | 24.66 | 27.48 | 12.86 |
| OAG-BERT | 5.59 | 8.37 | 11.11 | 9.62 | 13.09 | 16.39 | 28.49 | 32.48 | 35.97 | 17.90 |
| SciNCL | 24.91 | 36.57 | 46.25 | 34.84 | 48.50 | 58.77 | 31.65 | 39.30 | 45.84 | 40.74 |
| SPECTER 2.0 | 23.53 | 34.47 | 43.54 | 33.75 | 46.46 | 56.23 | 30.36 | 36.10 | 40.96 | 38.38 |
| SciMult-Vanilla | 32.61 | 46.33 | 56.40 | 38.23 | 52.08 | 62.02 | 41.61 | 50.20 | 56.94 | 48.49 |
| SciMult-MHAExpert | 34.81 | 49.31 | 59.71 | **39.82** | **54.16** | **64.50** | 42.33 | 51.25 | 58.25 | 50.46 |
| SciMult-FFNExpert | **36.34** | **51.31** | **61.97** | 38.12 | 52.49 | 62.91 | 43.36 | **52.31** | **59.05** | **50.87** |
| SciMult-Prefix | 34.18 | 49.27 | 60.52 | 37.70 | 52.21 | 62.85 | **43.54** | 52.28 | 59.00 | 50.17 |
| SciMult-Instruction | 33.58 | 47.94 | 58.37 | 36.22 | 49.98 | 60.26 | 43.37 | 52.10 | 58.72 | 48.95 |

Table 5: Link prediction performance on SciDocs (Cohan et al., 2020) and PMC-Patients (Zhao et al., 2022) under the retrieval setting.

| Patient-to-Article Retrieval | MRR | P@10 | nDCG@10 | R@1K |
|---|---|---|---|---|
| DPR (SciMult-MHAExpert) | **64.44** | **22.12** | **28.62** | **69.09** |
| DPR (PubMedBERT) | 42.96 | 16.08 | 19.51 | 63.40 |
| DPR (SPECTER) | 46.41 | 15.59 | 19.70 | 57.98 |
| DPR (BioLinkBERT) | 40.89 | 15.33 | 18.47 | 62.44 |
| BM25 | 48.22 | 9.97 | 15.28 | 30.64 |
| Contriever | 15.03 | 3.41 | 4.62 | 16.74 |
| SentBERT | 10.58 | 2.71 | 3.53 | 13.52 |

| Patient-to-Patient Retrieval | MRR | P@10 | nDCG@10 | R@1K |
|---|---|---|---|---|
| DPR (SciMult-MHAExpert) | **25.35** | **6.65** | **22.39** | **83.78** |
| BM25 | 22.86 | 4.67 | 18.29 | 69.66 |
| DPR (BioLinkBERT) | 21.20 | 5.59 | 18.06 | 80.49 |
| DPR (PubMedBERT) | 19.37 | 5.05 | 16.30 | 79.35 |
| DPR (SPECTER) | 15.08 | 3.79 | 12.27 | 73.01 |
| Contriever | 10.50 | 2.24 | 8.01 | 52.64 |
| SentBERT | 5.28 | 1.17 | 3.88 | 37.55 |

Table 6: Comparison between SciMult-MHAExpert and models on the leaderboard of PMC-Patients (Zhao et al., 2022) Patient-to-Article Retrieval and Patient-to-Patient Retrieval tasks. SciMult-MHAExpert achieves the new state-of-the-art performance.

supervised SciMult by further fine-tuning it on the provided training data. Specifically, we pick SciMult-MHAExpert (because of its overall good performance according to Table 9) and use DPR (Karpukhin et al., 2020) to fine-tune it. The comparison between DPR(SciMult-MHAExpert) and existing models on the leaderboard of PMC-Patients[4] is shown in Table 6. Our model outperforms all existing models and achieves the new state-of-the-art.

For the reranking setting, we follow the original evaluation protocol of SciDocs (Cohan et al., 2020): For each query paper $p_Q$, a small set of candidate papers $\{p_{C1}, p_{C2}, ...\}$ is given, which contains up to 5 positives and 25 random negatives. The task aims to rank the positives higher than the negatives, so we use MAP and nDCG as evaluation metrics. The Recommendation dataset (Kanakia et al., 2019) has a similar format, except that the score of each

[4] https://pmc-patients.github.io/

candidate is not binary and ranges from 1 to 5. As a result, the model needs to rank candidate papers with higher scores in front of those with lower scores, and we use nDCG and nDCG@$k$ ($k = 5, 10$) as the metrics. (MAP cannot be used in Recommendation because relevance is not binary.) The performance is demonstrated in Table 7.

From Table 7, we observe that: (1) On Sci-Docs, SciMult variants outperform SPECTER in most cases. Note that for the link prediction task, SciMult uses exactly the same pre-training data (including hard negatives) as SPECTER (Cohan et al., 2020). Therefore, this observation implies that exploiting pre-training data from other tasks can benefit the link prediction performance, which validates our motivation to develop a multi-task learning framework. (2) On SciDocs, SciMult variants can rarely outperform SciNCL and SPECTER 2.0 (but the gaps are not evident). This is possibly because SciNCL uses more complicated hard negative sampling strategies, and SPECTER 2.0 exploits more diverse pre-training data and tasks. In fact, Ostendorff et al. (2022) have pointed out the data leakage issue that 40.5% of SciDocs papers appear in the pre-training data. This motivates us to incorporate the cross-domain Recommendation dataset, on which SciMult-Vanilla and SciMult-MHAExpert consistently outperform all baselines.

## 4.6 Search

We use nDCG@10, the primary metric of BEIR (Thakur et al., 2021), to measure models' performance on the search task. The results are shown in Table 8. We find that, on the four cross-domain evaluation datasets and on average, four SciMult variants can outperform all baselines. In particular, on NFCorpus, only SciMult can beat BM25, while

| Link Prediction (Reranking) | SciDocs (Cohan et al., 2020) | | | | | | | | Kanakia et al. (2019) | | | |
| | Co-view | | Co-read | | Cite | | Co-cite | | Recommendation | | | Average |
| | MAP | nDCG | MAP | nDCG | MAP | nDCG | MAP | nDCG | nDCG@5 | nDCG@10 | nDCG | |
|---|---|---|---|---|---|---|---|---|---|---|---|---|
| Citeomatic (Bhagavatula et al., 2018) | 81.1† | 90.2† | 80.5† | 90.2† | 86.3† | 94.1† | 84.4† | 92.8† | – | – | – | – |
| Kanakia et al. (2019) | – | – | – | – | – | – | – | – | 83.88‡ | 87.71‡ | 93.59‡ | – |
| SciBERT (Beltagy et al., 2019) | 50.7† | 73.1† | 47.7† | 71.1† | 48.3† | 71.7† | 49.7† | 72.6† | 77.17 | 82.49 | 90.86 | 66.86 |
| SentBERT (Reimers and Gurevych, 2019) | 68.2† | 83.3† | 64.8† | 81.3† | 63.5† | 81.6† | 66.4† | 82.8† | 76.75 | 81.49 | 90.80 | 76.45 |
| SPECTER (Cohan et al., 2020) | 83.6† | 91.5† | 84.5† | 92.4† | 88.3† | 94.9† | 88.1† | 94.8† | 83.38 | 87.39 | 93.64 | 89.32 |
| PubMedBERT (Gu et al., 2021) | 59.43 | 78.23 | 55.59 | 75.63 | 51.81 | 73.43 | 58.19 | 77.80 | 77.30 | 82.21 | 91.09 | 70.97 |
| LinkBERT (Yasunaga et al., 2022) | 44.21 | 67.76 | 41.04 | 65.31 | 39.33 | 63.91 | 42.84 | 67.18 | 76.10 | 80.89 | 90.47 | 61.73 |
| BioLinkBERT (Yasunaga et al., 2022) | 56.46 | 76.38 | 50.76 | 72.18 | 47.73 | 70.55 | 52.94 | 74.44 | 77.02 | 81.78 | 90.73 | 68.27 |
| OAG-BERT (Liu et al., 2022) | 64.61 | 81.50 | 60.13 | 78.65 | 57.35 | 77.60 | 62.47 | 80.92 | 76.73 | 82.12 | 90.96 | 73.91 |
| SciNCL (Ostendorff et al., 2022) | **85.3†** | **92.3†** | **87.5†** | **93.9†** | 93.6† | 97.3† | **91.6†** | 96.4† | 85.33 | 88.38 | 94.34 | **91.45** |
| SPECTER 2.0 (Singh et al., 2022) | 85.18† | 92.27† | 86.95† | 93.53† | 92.23† | 96.84† | 91.13† | 96.28† | 86.03 | 89.12 | 94.59 | 91.29 |
| SciMult-Vanilla | 83.99 | 91.68 | 86.66 | 93.67 | 91.37 | 96.26 | 91.50 | 96.45 | **87.32** | 89.32 | **94.88** | 91.19 |
| SciMult-MHAExpert | 83.92 | 91.60 | 86.45 | 93.55 | 92.58 | 96.92 | 91.47 | 96.36 | 86.68 | **89.45** | 94.77 | 91.25 |
| SciMult-FFNExpert | 83.23 | 91.26 | 85.61 | 93.20 | 93.77 | 97.42 | 90.39 | 95.94 | 85.75 | 88.45 | 94.29 | 90.85 |
| SciMult-Prefix | 83.43 | 91.48 | 85.89 | 93.27 | **94.28** | **97.60** | 90.73 | 96.09 | 86.05 | 88.85 | 94.66 | 91.12 |
| SciMult-Instruction | 82.13 | 90.88 | 84.14 | 92.36 | 92.63 | 96.91 | 89.27 | 95.43 | 86.49 | 88.81 | 94.51 | 90.32 |

Table 7: Link prediction performance on SciDocs (Cohan et al., 2020) and Recommendation (Kanakia et al., 2019) under the reranking setting. Scores with † are reported in Cohan et al. (2020), Ostendorff et al. (2022), and Singh et al. (2022). Scores with ‡ are calculated from the model output released by Kanakia et al. (2019).

| Search | SciRepEval (Singh et al., 2022) | | BEIR (Thakur et al., 2021) | | | |
| | Search (2022) | TREC-COVID (2021) | TREC-COVID (2021) | SciFact (2020) | NFCorpus (2016) | |
| | nDCG@10 | nDCG@10 | nDCG@10 | nDCG@10 | nDCG@10 | Average |
|---|---|---|---|---|---|---|
| BM25 | 73.47 | 55.86 | 57.79 | 65.63 | 30.00 | 56.55 |
| SciBERT | 71.39 | 40.98 | 4.17 | 0.88 | 1.90 | 23.86 |
| SentBERT | 71.84 | 51.30 | 20.73 | 9.40 | 6.69 | 31.99 |
| SPECTER | 73.42 | 66.45 | 29.91 | 49.74 | 15.83 | 47.07 |
| PubMedBERT | 70.77 | 45.28 | 7.56 | 0.30 | 1.09 | 25.00 |
| LinkBERT | 71.66 | 52.45 | 2.28 | 0.49 | 1.77 | 25.73 |
| BioLinkBERT | 71.18 | 36.01 | 3.17 | 0.12 | 0.98 | 22.29 |
| OAG-BERT | 72.17 | 55.09 | 7.11 | 18.33 | 8.48 | 32.24 |
| SciNCL | 73.78 | 73.50 | 34.69 | 56.51 | 22.34 | 52.16 |
| SPECTER 2.0 | **78.22†** | 79.43 | 58.48 | 67.16 | 22.84 | 61.23 |
| SciMult-Vanilla | 76.44 | **86.76** | 67.22 | **70.76** | **31.20** | 66.48 |
| SciMult-MHAExpert | 76.33 | 86.29 | **71.18** | 70.67 | 30.79 | **67.05** |
| SciMult-FFNExpert | 76.02 | 82.32 | 52.15 | 63.57 | 27.48 | 60.31 |
| SciMult-Prefix | 76.55 | 82.83 | 68.15 | 70.70 | 30.02 | 65.65 |
| SciMult-Instruction | 75.86 | 83.59 | 61.05 | 70.62 | 30.25 | 64.27 |

| | MHAExpert | FFNExpert | Prefix | Instruction |
|---|---|---|---|---|
| Classification (Fine) Table 3 | +3.15% | -0.95% | -0.51% | +0.68% |
| Classification (Coarse) Table 4 | +1.02% | +0.33% | +1.50% | +0.93% |
| Link Prediction (Retrieval) Table 5 | +4.06% | +4.91% | +3.46% | +0.95% |
| Link Prediction (Reranking) Table 7 | +0.07% | -0.37% | -0.08% | -0.95% |
| Search Table 8 | +0.86% | -9.28% | -1.25% | -3.32% |

Table 8: Search performance on SciRepEval-Search (Singh et al., 2022), TREC-COVID (Voorhees et al., 2021), SciFact (Wadden et al., 2020), and NFCorpus (Boteva et al., 2016). The score with † is reported in Singh et al. (2022).

Table 9: Relative performance change of different SciMult variants in comparison with SciMult-Vanilla in terms of the average evaluation metric.

all baseline LMs are lacking by a clear margin.

## 4.7 Overall Analysis

To summarize, in Tables 3, 4, 5, and 8, all SciMult variants except SciMult-FFNExpert can always beat all baselines in terms of the average metric.

Meanwhile, to systematically examine whether our proposed techniques can mitigate task interference, we calculate the relative performance change of the four non-Vanilla variants in comparison with SciMult-Vanilla in each task. Table 9 shows the results. We observe that SciMult-MHAExpert improves SciMult-Vanilla across all tasks, which implies that the MoE architecture with task-specific MHA sub-layers effectively overcome task interference during multi-task pre-training. By contrast, other proposed techniques are advantageous in a subset of tasks, such as coarse-grained classification and link prediction under the retrieval setting.

To validate the design choices of SciMult, we conduct more analysis through controlled experiments, which can be found in Appendix E. To briefly summarize, we observe that: (1) Using hard negatives in multi-task contrastive learning helps produce higher-quality text representations in general and benefit all tasks. (2) When pre-training non-Vanilla variants, warming up the LM by training a Vanilla variant during initial steps yields better performance. (3) Using some other reasonable instructions during inference does not significantly affect model performance. (4) Although label definitions are used for paper classification during pre-training and are beneficial to the classification performance during inference, our model can still outperform baselines in classification by taking label names as the only input.

## 5 Related Work

**Scientific Literature Understanding.** Recent LMs pre-trained on domain-specific scientific texts,

from Transformer-based ones such as SciBERT (Beltagy et al., 2019), BioBERT (Lee et al., 2020), ChemBERT (Guo et al., 2021), and PubMed-BERT (Gu et al., 2021) to GPT-based ones such as SciGPT2 (Luu et al., 2021) and BioGPT (Luo et al., 2022), aim to learn high-quality contextualized text representations for scientific literature understanding. Subsequent studies have utilized these LMs to fine-grained paper classification (Zhang et al., 2022, 2023a), cite-worthiness detection (Wright and Augenstein, 2021), scientific claim verification (Wadden et al., 2020), and so on.

Besides text information, metadata associated with scientific papers are also broadly considered. For example, Citeomatic (Bhagavatula et al., 2018), SPECTER (Cohan et al., 2020), BioLinkBERT (Yasunaga et al., 2022), and SciNCL (Ostendorff et al., 2022) leverage citation links between papers; OAG-BERT (Liu et al., 2022) models venues, authors, fields-of-study, and affiliations during LM pre-training; S2AND (Subramanian et al., 2021) further utilizes year, email, and position information for author name disambiguation. Nevertheless, all aforementioned models either consider typical LM pre-training tasks (*e.g.,* MLM and NSP) only or focus on one additional task during model training (*e.g.,* citation prediction). In comparison, `SciMult` exploits data from heterogeneous sources and proposes a multi-task learning framework that can be applied to a wide range of tasks.

**Contrastive Learning and Multi-task Learning in the Scientific Domain.** SPECTER (Cohan et al., 2020) and SciNCL (Ostendorff et al., 2022) are pioneering studies on using contrastive learning to enhance scientific LMs. They propose to derive positive and negative contrastive pairs from citation triplets and demonstrate the power of mining hard negatives. MICoL (Zhang et al., 2022) and CitationSum (Luo et al., 2023) adopt contrastive learning to multi-label classification and summarization of scientific papers, respectively. As for multi-task learning, Luan et al. (2018) propose a multi-task scientific knowledge graph construction framework by jointly identifying entities, relations, and coreference; Wang et al. (2019) treat multiple biomedical named entity recognition datasets (with different types of entities annotated) as multiple tasks so that they can mutually benefit each other. However, these studies do not have specific designs to tackle task interference. To the best of our knowledge, the recent work by Singh et al. (2022) is the most relevant one to `SciMult`, which pre-trains a scientific LM on various tasks and uses adapters

(Houlsby et al., 2019) and control codes (Keskar et al., 2019) to produce task-aware paper representations. We believe our study is orthogonal to Singh et al. (2022) as the techniques considered by us, including the Mixture-of-Experts Transformer (Fedus et al., 2022) and instruction tuning (Wei et al., 2022), are distinct from theirs.

# 6 Conclusions

In this work, we propose to pre-train scientific LMs via multi-task contrastive learning. To mitigate task interference, we adopt two strategies: Task-aware specialization considers a Mixture-of-Experts Transformer architecture so that each task has its unique components, while instruction tuning relies on task-specific instructions to produce task-aware text representations. Extensive experiments on a comprehensive collection of benchmark datasets demonstrate the advantages of our models against competitive scientific LMs in extreme multi-label classification, link prediction, and search. In particular, we achieve the new state-of-the-art performance on the PMC-Patients leaderboard. We also show that task-specific MHA sub-layers are beneficial to the model performance across all examined tasks, whereas the benefit of other proposed techniques is not consistent. Further analysis validates some of our design choices, such as hard negative mining in extreme classification.

# Limitations

In comparison with previous studies on instruction tuning considering tens of (Asai et al., 2023) to thousands of (Wang et al., 2022) tasks, we do not explore that many different scientific literature understanding tasks. As a result, we may not unleash the power of instruction tuning to the utmost extent. It is of our interest to collect more datasets in the scientific domain, such as author name disambiguation (Subramanian et al., 2021) and biomedical question answering (Jin et al., 2019), and then explore whether the instruction tuning model trained on more tasks can mitigate task inference and support zero-shot transfer to new tasks. Also, our study focuses on the Bi-Encoder architecture only. It would be meaningful to investigate other types of architectures, such as Cross-Encoders and late-interaction models (Humeau et al., 2020; Khattab and Zaharia, 2020), and study how to apply task-aware specialization and instruction tuning to them.

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

# A Dataset Details

Statistics of the pre-training and evaluation datasets are summarized in Table A1 and Table A2, respectively.

We lowercase the text in all datasets.

## A.1 Classification

The MAPLE benchmark (Zhang et al., 2023b) is available at `https://github.com/yuzhimanhua/MAPLE`. The SciDocs benchmark (Cohan et al., 2020) is available at `https://github.com/allenai/scidocs`.

**MAPLE (CS-Journal).** Because label definitions are not released in MAPLE, we adopt the definitions of 15,808 CS fields-of-study in Zhang et al. (2022). This label space is not identical to the original label space of CS-Journal (with 15,540 labels), and we remove papers that do not have any relevant labels in the adopted label space. Because some general labels (*e.g.,* "`Machine Learning`") are relevant to a large proportion of papers in the original dataset, we perform downsampling to ensure each label $l$ appears in at most 100 positive $(p, l)$ pairs in the pre-training data. In this way, the LM will not be overwhelmed by cases of general labels and can learn more semantics of specific labels (*e.g.,* "`Lagrangian Support Vector Machine`").

**MAPLE (Biology-MeSH and Medicine-MeSH).** We combine these two datasets and use a shared label space of 30,194 MeSH terms in the 2022 version of MeSH. The definition of each MeSH term is its "Scope Note". Same as above, we perform downsampling to ensure each label $l$ appears in at most 100 positive $(p, l)$ pairs in the pre-training data.

**MAPLE (Coarse).** There are 19 fields in MAPLE. For each of the 18 non-CS fields, we sample 50,000 papers, label it as the corresponding coarse field (*e.g.,* a paper $p$ from the Mathematics field will form the paper-label pair $(p, \text{Mathematics})$), and put it into the pre-training data. These data can train the LM to perform coarse-grained classification.

**MAPLE (CS-Conference).** The candidate label space of CS-Conference for evaluation is the same as that of CS-Journal for pre-training (*i.e.,* 15,808 labels from Zhang et al. (2022)). Because this label space is not identical to the original label space of CS-Conference (with 13,613 labels), we remove papers that do not have any relevant labels in our adopted label space.

**MAPLE (Chemistry-MeSH).** The candidate label space of Chemistry-MeSH for evaluation is the same as that of Biology-MeSH and Medicine-MeSH during pre-training, which is a superset of the original label space of Chemistry-MeSH.

**MAPLE (Geography and Psychology).** We directly use the original data. Since we do not have label definitions in these two fields, only label names are used as the text information.

**SciDocs (MAG Fields and MeSH Diseases).** Under the *Linear Classifier* setting, we directly use the original data and their train-val-test split. Under the *Bi-Encoder* setting, because no training and validation samples are needed, we merge the training, validation, and testing sets for evaluation.

## A.2 Link Prediction

The SciDocs benchmark (Cohan et al., 2020) is available at `https://github.com/allenai/scidocs`. The PMC-Patients dataset (Zhao et al., 2022) is available at `https://github.com/pmc-patients/pmc-patients`. The Recommendation dataset (Kanakia et al., 2019) is available at `https://github.com/akanakia/microsoft-academic-paper-recommender-user-study`.

**Citation Prediction Triplets.** We directly use the data from `https://huggingface.co/datasets/allenai/scirepeval/viewer/cite_prediction`. In each triplet $(p_Q, p_{C+}, p_{C-})$, $p_Q$ cites $p_{C+}$ and $p_Q$ does not cite $p_{C-}$. With a probability of about 40%, $p_{C-}$ is a hard negative.

**SciDocs (Co-view, Co-read, Cite, and Co-cite).** Under the *reranking* setting, we directly use the original data. Under the *retrieval* setting, for "Cite" links, a query-candidate paper pair $(p_Q, p_C)$ is viewed as positive if $p_Q$ cites $p_C$ according to the dataset; for "Co-cite" links, a query-candidate pair $(p_Q, p_C)$ is viewed as positive if $p_Q$ and $p_C$ are co-cited by at least 10 papers in the dataset; for "Co-view" and "Co-read" links, we cannot derive a version for retrieval because we do not know whether $(p_Q, p_C)$ is positive when $p_C$ is not in the reranking candidate pool of $p_Q$ in the original data.

**PMC-Patients.** In Table 5, because no training and validation samples are needed, we merge the training, validation, and testing sets of the Patient-to-Patient Retrieval task for evaluation. In Table 6, to compare `SciMult` with other models on the PMC-Patients leaderboard, we adopt the original train-val-test split.

| Dataset | #Queries | #Positive (Query, Candidate) Pairs |
|---|---|---|
| **Classification** | | |
| MAPLE (CS-Journal) (Zhang et al., 2023b) | 358,447 | 1,242,885 |
| MAPLE (Biology-MeSH + Medicine-MeSH) (Zhang et al., 2023b) | 1,345,128 | 1,976,858 |
| MAPLE (Coarse) (Zhang et al., 2023b) | 900,000 | 900,000 |
| **Link Prediction** | | |
| Citation Prediction Triplets (Cohan et al., 2020) | 165,340 | 819,836 |
| **Search** | | |
| SciRepEval-Search (Singh et al., 2022) | 528,497 | 620,033 |

Table A1: Statistics of Pre-training Data.

| Dataset | #Queries | #Candidates |
|---|---|---|
| **Fine-grained Classification** | | |
| MAPLE (CS-Conference) (Zhang et al., 2023b) | 261,781 | 15,808 |
| MAPLE (Chemistry-MeSH) (Zhang et al., 2023b) | 762,129 | 30,194 |
| MAPLE (Geography) (Zhang et al., 2023b) | 73,883 | 3,285 |
| MAPLE (Psychology) (Zhang et al., 2023b) | 372,954 | 7,641 |
| **Coarse-grained Classification** | | |
| SciDocs (MAG Fields) (Cohan et al., 2020) | 25,001 | 19 |
| SciDocs (MeSH Diseases) (Cohan et al., 2020) | 23,473 | 11 |
| **Link Prediction (Retrieval)** | | |
| SciDocs (Cite) (Cohan et al., 2020) | 92,214 | 142,009 |
| SciDocs (Co-cite) (Cohan et al., 2020) | 54,543 | 142,009 |
| PMC-Patients (Patient-to-Patient Retrieval, Zero-shot) (Zhao et al., 2022) | 100,327 | 155,151 |
| PMC-Patients (Patient-to-Article Retrieval, Supervised) (Zhao et al., 2022) | 5,959 | 1,413,087 |
| PMC-Patients (Patient-to-Patient Retrieval, Supervised) (Zhao et al., 2022) | 2,812 | 155,151 |
| **Link Prediction (Reranking)** | | |
| SciDocs (Co-view) (Cohan et al., 2020) | 1,000 | reranking, 29.98 for each query on average |
| SciDocs (Co-read) (Cohan et al., 2020) | 1,000 | reranking, 29.98 for each query on average |
| SciDocs (Cite) (Cohan et al., 2020) | 1,000 | reranking, 29.93 for each query on average |
| SciDocs (Co-cite) (Cohan et al., 2020) | 1,000 | reranking, 29.95 for each query on average |
| Recommendation (Kanakia et al., 2019) | 137 | reranking, 16.28 for each query on average |
| **Search** | | |
| SciRepEval-Search (Singh et al., 2022) | 2,637 | reranking, 10.00 for each query on average |
| TREC-COVID in SciRepEval (Voorhees et al., 2021) | 50 | reranking, 1386.36 for each query on average |
| TREC-COVID in BEIR (Voorhees et al., 2021) | 50 | 171,332 |
| SciFact (Wadden et al., 2020) | 1,109 | 5,183 |
| NFCorpus (Boteva et al., 2016) | 3,237 | 3,633 |

Table A2: Statistics of Evaluation Datasets.

**Recommendation.** In the original dataset, each paper only has its MAG ID (Sinha et al., 2015), while its title and abstract are not included. We find the title and abstract of most papers in a version of the Microsoft Academic Graph downloaded in 2021. Those papers whose titles and abstracts are not found are removed from evaluation.

### A.3 Search

The SciRepEval benchmark (Singh et al., 2022) is available at `https://github.com/allenai/scirepeval`. The BEIR benchmark (Thakur et al., 2021) is available at `https://github.com/beir-cellar/beir`.

**SciRepEval-Search.** The original data scores each query-paper pair $(q, p)$ in the range of 0 to 14 according to user click-through events from a scholarly search engine. The training and validation sets of SciRepEval-Search are put into our pre-training

data, where we treat all $(q, p)$ pairs with a positive score as positive $(q, p)$ pairs. The testing set of SciRepEval-Search is utilized for in-domain evaluation.

**TREC-COVID.** We directly use the original testing set. In the SciRepEval version, each query has multiple segments separated by [SEP].

**SciFact and NFCorpus.** We directly use the original data. Because no training and validation samples are needed, we merge the training, validation, and testing sets for evaluation.

## B  Baseline Details

• **SciBERT** (Beltagy et al., 2019) is an LM pretrained on scientific text using masked language modeling (MLM) and next sentence prediction (NSP).

• **SentBERT** (Reimers and Gurevych, 2019) is a

general-domain LM that leverages negative sampling to fine-tune BERT for producing better sentence embeddings.

- **SPECTER** (Cohan et al., 2020) uses paper citations to generate positive and negative samples for contrastive fine-tuning of SciBERT.

- **PubMedBERT** (Gu et al., 2021) is a biomedical LM pre-trained on PubMed papers using MLM and NSP. We use the checkpoint pre-trained on abstracts rather than that on full texts because the former one performs better on the majority of our evaluation tasks.

- **LinkBERT** and **BioLinkBERT** (Yasunaga et al., 2022) leverage a Cross-Encoder architecture that concatenates two linked text segments together and are trained through MLM and NSP on Wikipedia and PubMed, respectively.

- **OAG-BERT** (Liu et al., 2022) is an entity-augmented scientific LM pre-trained on both academic texts and their associated metadata entities (*e.g.,* venues, authors) through masked entity prediction.

- **SciNCL** (Ostendorff et al., 2022) advances the sampling strategy of SPECTER to create higher-quality positives and negatives for neighborhood contrastive learning.

- **SPECTER 2.0** (Singh et al., 2022) is the successor to SPECTER pre-trained on a much larger collection of citation prediction triplets and more diverse tasks from the SciRepEval benchmark (Singh et al., 2022). We adopt SPECTER 2.0-Adapters to generate task-specific embeddings for different tasks.

For all baselines, we set the similarity function as $\text{sim}(q, c) = \cos(\boldsymbol{q}, \boldsymbol{c})$ where $\boldsymbol{q}$ and $\boldsymbol{c}$ are query and candidate embeddings, respectively, after LM encoding. (Except for reranking tasks on SciDocs, where the evaluation code explicitly sets $\text{sim}(q, c) = -||\boldsymbol{q} - \boldsymbol{c}||_2$.)

When using SentBERT and OAG-BERT, we take the average of all token embeddings to represent the entire input sequence because this leads to significantly better performance; when using other baselines above, we take the [CLS] embedding.

For SPECTER 2.0, we try to use the most proper variant in each task. To be specific, following Singh et al. (2022), for coarse-grained classification, we

use the classification adapter; for link prediction, we use the proximity adapter; for search, we use the adhoc query adapter to encoder queries and the proximity adapter to encode candidate papers. For fine-grained classification, we test both the classification adapter and the proximity adapter, the latter of which achieves better performance on average, so we choose the proximity adapter.

## C  Hyperparameter Configurations of `SciMult`

For pre-training, we use the AdamW optimizer (Loshchilov and Hutter, 2019) with $(\beta_1, \beta_2) = (0.9, 0.999)$ and warm up the learning rate for the first 5% of the steps. We train the model for 20 epochs with a peak learning rate of 3e-4 and a weight decay of 0.01. The training is on 48 V100 GPUs with fp16, and the batch size per GPU is 32. When pre-training the four non-Vanilla model variants, we adopt a two-stage strategy: We first train a `SciMult`-Vanilla checkpoint to learn common knowledge shared across different tasks. Then, starting from that checkpoint, we leverage task-aware specialization or instruction tuning for continual pre-training, expecting the model to learn task-specific skills.

## D  More Results on Fine-grained Classification

In this section, we show the fine-grained classification performance of compared methods on MAPLE (Zhang et al., 2023b) without the label name matching heuristic. The results are in Table A3. We find that: (1) In comparison with the results in Table 3, after removing the label name matching heuristic, all models perform consistently worse. This observation validates the effectiveness of the heuristic proposed in Zhang et al. (2023b). (2) Most findings drawn from Table 3 still hold in Table A3. For example, all `SciMult` variants can outperform all baselines in terms of the average metric; `SciMult`-MHAExpert is always the best on cross-domain evaluation datasets.

## E  Analysis

In this section, we analyze some design choices of `SciMult` through controlled experiments.

### E.1  Effect of Hard Negatives

We first demonstrate the contribution of hard negatives in our contrastive learning framework. Since

| Fine-grained classification | MAPLE (Zhang et al., 2023b) | | | | | | | | | | | |
| | CS-Conference | | | Chemistry-MeSH | | | Geography | | | Psychology | | | Average |
| | R@20 | R@50 | R@100 | R@20 | R@50 | R@100 | R@20 | R@50 | R@100 | R@20 | R@50 | R@100 | |
|---|---|---|---|---|---|---|---|---|---|---|---|---|---|
| BM25 (Robertson and Walker, 1994) | 17.41 | 23.73 | 29.12 | 7.78 | 10.75 | 13.41 | 15.81 | 24.15 | 35.99 | 13.00 | 18.45 | 24.90 | 19.54 |
| SciBERT (Beltagy et al., 2019) | 0.84 | 1.98 | 3.63 | 0.77 | 1.58 | 2.71 | 3.73 | 8.05 | 14.08 | 1.72 | 3.48 | 5.84 | 4.03 |
| SentBERT (Reimers and Gurevych, 2019) | 2.94 | 5.24 | 7.97 | 1.20 | 2.21 | 3.45 | 7.68 | 14.26 | 20.97 | 2.62 | 4.79 | 7.67 | 6.75 |
| SPECTER (Cohan et al., 2020) | 16.28 | 24.50 | 32.60 | 11.80 | 17.94 | 23.88 | 18.12 | 28.02 | 37.36 | 15.18 | 23.46 | 32.24 | 23.45 |
| PubMedBERT (Gu et al., 2021) | 0.87 | 1.68 | 2.75 | 0.80 | 1.28 | 1.83 | 4.51 | 8.69 | 12.56 | 2.92 | 5.95 | 9.79 | 4.47 |
| LinkBERT (Yasunaga et al., 2022) | 1.33 | 2.55 | 4.14 | 0.68 | 1.58 | 2.72 | 0.57 | 1.16 | 2.34 | 0.65 | 1.13 | 1.72 | 1.71 |
| BioLinkBERT (Yasunaga et al., 2022) | 1.02 | 1.95 | 3.05 | 0.46 | 0.87 | 1.40 | 0.32 | 0.68 | 1.40 | 0.14 | 0.39 | 0.79 | 1.04 |
| OAG-BERT (Liu et al., 2022) | 2.61 | 4.24 | 6.09 | 1.72 | 2.68 | 3.74 | 1.96 | 3.22 | 4.75 | 0.79 | 1.31 | 1.97 | 2.92 |
| SciNCL (Ostendorff et al., 2022) | 17.42 | 25.32 | 32.82 | 13.96 | 20.33 | 26.22 | 21.80 | 32.96 | 43.02 | 22.76 | 32.85 | 42.30 | 27.65 |
| SPECTER 2.0 (Singh et al., 2022) | 20.90 | 30.21 | 38.87 | 18.72 | 27.08 | 34.41 | 31.61 | 45.26 | 56.85 | 26.27 | 38.85 | 50.21 | 34.94 |
| SciMult-Vanilla | 31.52 | 46.04 | 58.61 | **26.96** | **39.63** | **50.37** | 32.20 | 46.96 | 59.12 | 30.02 | 40.48 | 50.31 | 42.69 |
| SciMult-MHAExpert | 35.63 | 51.00 | 64.02 | 26.11 | 38.85 | 49.89 | **37.70** | **52.44** | **63.99** | **32.69** | **44.80** | **56.28** | **46.12** |
| SciMult-FFNExpert | 38.43 | 52.66 | 64.89 | 22.83 | 34.93 | 46.05 | 26.35 | 41.37 | 55.03 | 27.95 | 40.05 | 51.23 | 41.81 |
| SciMult-Prefix | 38.42 | 52.58 | 64.51 | 22.67 | 34.88 | 45.95 | 28.25 | 43.00 | 56.55 | 26.09 | 38.48 | 49.71 | 41.76 |
| SciMult-Instruction | **38.47** | **52.88** | **65.10** | 24.36 | 36.91 | 48.19 | 29.55 | 43.91 | 57.40 | 28.80 | 41.71 | 53.66 | 43.41 |

Table A3: Fine-grained classification performance on MAPLE (Zhang et al., 2023b) when the compared methods do not use the "label name matching" heuristic.

| | Vanilla (w/ Hard Negative) | Vanilla (w/o Hard Negative) |
|---|---|---|
| Classification (Fine) | **60.31** | 59.89 |
| Classification (Coarse) | 75.51 | **77.44** |
| Link Prediction (Retrieval) | **48.49** | 45.90 |
| Link Prediction (Reranking) | **91.19** | 90.88 |
| Search | **66.48** | 65.87 |

Table A4: Average metrics of SciMult-Vanilla with and without hard negatives in each evaluation task.

| | MHAExpert (w/ Warm-up) | MHAExpert (w/o Warm-up) |
|---|---|---|
| Classification (Fine) | **62.21** | 59.98 |
| Classification (Coarse) | **76.28** | 76.08 |
| Link Prediction (Retrieval) | **50.46** | 47.59 |
| Link Prediction (Reranking) | **91.25** | 90.89 |
| Search | **67.05** | 62.34 |

Table A5: Average metrics of SciMult-MHAExpert with different pre-training strategies in each evaluation task.

the benefit of using hard negatives in citation prediction has been reported in Cohan et al. (2020) and Ostendorff et al. (2022), we mainly show how hard negative mining in fine-grained classification (proposed in subsection 3.4) improves the performance. Table A4 shows the average metrics of SciMult-Vanilla when trained with hard negatives of classification and without them (but still with hard negatives of the other tasks). We find that SciMult-Vanilla (with Hard Negative) outperforms SciMult-Vanilla (without Hard Negative) in most tasks, including not only classification but also link prediction and search. This observation indicates that the general quality of the learned text representations is enhanced after utilizing hard negatives of classification.

### E.2 Effect of the Pre-training Strategy

As mentioned in Appendix C, when pre-training non-Vanilla variants of SciMult, we follow a two-stage strategy: We first warm up the LM by training a Vanilla variant and then apply task-aware specialization or instruction tuning, hoping the first stage learns common knowledge and the second stage focuses on task-specific skills. We now explore the effect of this strategy by comparing

SciMult-MHAExpert with an ablation version: We skip the warm-up stage and directly use Pub-MedBERT as the initial checkpoint to start the second stage. Table A5 demonstrates the performance of SciMult-MHAExpert (with Warm-up) and SciMult-MHAExpert (without Warm-up). We find that the warm-up strategy has a positive contribution across all tasks.

### E.3 Effect of Instructions

We now explore the effect of using different instructions during inference. As shown in Table 1, during pre-training, we use the following instruction for the link prediction task:

"Find a pair of scientific papers that one paper cites the other."

We call it the 'Cite' instruction. Different from the pre-training data in which links are always citation links, the evaluation datasets consist of other link types. Therefore, we consider the following instructions:

"Find a pair of scientific papers that are co-viewed frequently."

"Find a pair of scientific papers that

| | "Co-view" Instruction | "Co-read" Instruction | "Cite" Instruction | "Co-cite" Instruction |
|---|---|---|---|---|
| Co-view Prediction | 81.16 | 81.47 | **82.13** | 82.08 |
| Co-read Prediction | 83.38 | 83.63 | 84.14 | **84.29** |
| Cite Prediction | 92.47 | 92.78 | 92.63 | **93.11** |
| Co-cite Prediction | 87.49 | 88.05 | **89.27** | 88.83 |

Table A6: MAP scores of `SciMult-Instruction` with different instructions in the SciDocs (Cohan et al., 2020) link prediction task under the reranking setting.

| | "Cite" Instruction | "Co-cite" Instruction | "Related" Instruction |
|---|---|---|---|
| Link Prediction (Retrieval) | 48.95 | 49.35 | 49.28 |
| Link Prediction (Reranking) | 90.32 | 90.31 | 90.35 |

Table A7: Average metrics of `SciMult-Instruction` with different instructions in link prediction tasks.

are co-read frequently."

"Find a pair of scientific papers that are co-cited frequently."

"Find a pair of scientific papers that one paper is related to the other."

We call these four instructions "Co-view", "Co-read", "Co-cite", and "Related", respectively.

Table A6 shows the MAP scores of `SciMult-Instruction` with different instructions in the Sci-Docs (Cohan et al., 2020) link prediction task under the reranking setting. We find that the "Co-view" and "Co-read" instructions underperform the "Cite" and "Co-cite" instructions in most cases, even in Co-view and Co-read link prediction.

Table A7 shows the average metrics of `SciMult-Instruction` with different instructions in link prediction tasks. We find that the "Cite", "Co-cite", and "Related" instructions are on par with each other.

### E.4 Availability of Label Definitions in Classification

Although both label names and definitions are used for paper classification during the pre-training of `SciMult`, we would like to emphasize that label definitions are not required but optional during inference. For example, as mentioned in Appendix A, the Geography and Psychology datasets do not have label definitions, thus classification on these two datasets relies on label names only. According to Table 3 and Table A3, `SciMult-MHAExpert` outperforms all baselines consistently on Geography

| CS-Conference | R@20 | R@50 | R@100 |
|---|---|---|---|
| SPECTER 2.0 (Name Only) | 48.14 | 53.93 | 58.98 |
| SPECTER 2.0 (Name+Definition) | 48.63 | 55.09 | 60.68 |
| SciMult-MHAExpert (Name Only) | 52.57 | 61.27 | 67.84 |
| SciMult-MHAExpert (Name+Definition) | 54.02 | 65.49 | 75.07 |
| **Chemistry-MeSH** | **R@20** | **R@50** | **R@100** |
| SPECTER 2.0 (Name Only) | 35.73 | 42.19 | 46.96 |
| SPECTER 2.0 (Name+Definition) | 36.17 | 43.06 | 48.26 |
| SciMult-MHAExpert (Name Only) | 37.83 | 46.47 | 52.59 |
| SciMult-MHAExpert (Name+Definition) | 39.41 | 50.92 | 59.59 |

Table A8: Fine-grained classification performance of SPECTER 2.0 and `SciMult-MHAExpert` when taking label names only or label names+definitions as input.

and Psychology.

On CS-Conference and Chemistry-MeSH, we conduct additional experiments by removing label definitions and letting the model use label names only during inference. The performance of `SciMult-MHAExpert` and SPECTER 2.0 (the strongest baseline in Table 3) is demonstrated in Table A8. We can observe that, after removing label definitions, although the performance of both SPECTER 2.0 and `SciMult-MHAExpert` drops (which is intuitive because we remove some useful information), `SciMult-MHAExpert` consistently performs better than SPECTER 2.0. In fact, even if SPECTER 2.0 uses label definitions, `SciMult-MHAExpert` still outperforms SPECTER 2.0 with label names only.