# OpenReview forum: "Pre-training Multi-task Contrastive Learning Models for Scientific Literature Understanding"
_EMNLP/2023/Conference — EMNLP 2023 Findings_

### Official Review · Reviewer_BsFu · 2023-08-04

**Soundness:** 3

**Excitement:**

3: Ambivalent: It has merits (e.g., it reports state-of-the-art results, the idea is nice), but there are key weaknesses (e.g., it describes incremental work), and it can significantly benefit from another round of revision. However, I won't object to accepting it if my co-reviewers champion it.

**Paper Topic And Main Contributions:**

This manuscript proposes a multi-task contrastive learning based language model pretraining framework for scientific literature understanding tasks, namely SciMult, which explores both task-aware specialization and instruction tuning, focusing on facilitating common knowledge sharing across different tasks. Extensive experiments on a comprehensive collection of benchmark datasets verify the effectiveness of the task-aware specialization strategy, where outperform state-of-the-art scientific LMs.

**Questions For The Authors:**

A. Better to clarify the differences between this manuscript and SciRepEval. The contributions of this manuscirpt seems limited.

**Reasons To Accept:**

A. It proposes four model variants to produce task-aware representations in order to learn common knowledge sharing across diverse tasks while preventing task-specific skills from interfering with each other.

B. Extensive experiments verify the effectiveness of SciMult, especially the task-aware specialization strategy, which outperforms state-of-the-art scientific LMs in both in-domain and corss-domain tasks.

C. The manuscript is easy to follow.

**Reasons To Reject:**

A. The motivation is not strong enough, although this manuscript notes that its techniques are distinct from the recent work SciRepEval[1] which shares the most relevant idea with SciMult. (Note that SciRepEval is first submitted 23 November, 2022.)

(1) Both of them proposed multi-task pretrained model for scientific literature understanding, while SciRepEval explore more scientific literature understanding tasks than SciMult.

(2) The pretraining strategies adopted by these two works are almost the same except specific model details, including task-aware adapters and instruction tuning, which aim to obtain task-specific representation. Although SciMult generalizes hard negatives to other tasks, which is of significant importance in citation prediction, the innovation is not that strong.

(3) Both of them consider not only in-domain but also cross-domain evaluationon different tasks and datasets.

[1] Amanpreet Singh, Mike D’Arcy, Arman Cohan, Doug Downey, and Sergey Feldman. 2022. Scirepeval: A multi-format benchmark for scientific document representations. arXiv preprint arXiv:2211.13308.

**Reproducibility:**

4: Could mostly reproduce the results, but there may be some variation because of sample variance or minor variations in their interpretation of the protocol or method.

**Reviewer Confidence:**

4: Quite sure. I tried to check the important points carefully. It's unlikely, though conceivable, that I missed something that should affect my ratings.

---

> ### Author Rebuttal · Authors · 2023-08-29
>
> Thanks for your thoughtful review and feedback! We discuss the raised points as follows:
>
> 1. **Comparison between SciMult and SciRepEval/SPECTER 2.0**
>
> We fully acknowledge that the SciRepEval/SPECTER 2.0 paper [1] is earlier than our submission. As evidence, in our submission, we cite the paper, compare our SciMult model with SPECTER 2.0 in all tasks, and discuss the difference between SciMult and SPECTER 2.0 in our Related Work section (Lines 611-621, Page 8). We would like to emphasize the differences in more detail here from three aspects.
>
> (1) **High-level focus**: This is reflected in the titles of the two papers. We focus on an empirical study of multiple techniques for mitigating task interference and a more performant scientific language model across the board for scientific understanding tasks. By contrast, the SciRepEval paper emphasizes its contribution to constructing a novel and comprehensive benchmark for scientific document representations in its title.
>
> (2) **Techniques**: This is already mentioned by the reviewer. To be specific, SPECTER 2.0 uses adapters and control codes, while SciMult uses Mixture-of-Experts Transformers and instruction tuning. Although the general goal of all these techniques is to produce task-specific embeddings, we argue that the two papers propose significantly different approaches to implementing this general goal. We do not think that one paper should be viewed as lacking motivation simply because it shares a general goal with another paper, especially when the goal of producing task-specific embeddings is common in multi-task learning (e.g., [2, 3]).
>
> (3) **Empirical performance**: We conduct a head-to-head performance comparison between SciMult and SPECTER 2.0 across various tasks. Experimental results show that SciMult outperforms SPECTER 2.0 in most cases (Tables 3, 4, 5, 7, and A4). The exception happens on the SciDocs dataset, where SciMult-MHAExpert is on par with or slightly underperforms SPECTER 2.0. However, we need to mention the data leakage problem of SciDocs pointed out by Ostendorff et al. in [4]: “**40.5%** of SciDoc’s papers leaking into SPECTER’s training data”. Because SPECTER, SciNCL, SPECTER 2.0, and SciMult all use SPECTER’s training data (i.e., the 819K citation prediction triplets) during pre-training, it becomes questionable how indicative the performance comparison can be if SciDocs is used for evaluation. In comparison, in the PMC-Patients dataset we use, **none** of the texts in the Patient-to-Patient task and only **0.2%** of the papers in the Patient-to-Article task can be found in SPECTER’s training data.
>
> (4) **Datasets**: We agree with the reviewer that SciRepEval has a wider range of tasks, some of which (e.g., regression, author name disambiguation) are not explored in SciMult. However, there are also tasks and datasets studied in SciMult but not in SciRepEval, such as PMC-Patients, Recommendation, SciFact, and NFCorpus. In particular, SciMult-MHAExpert achieves the new state-of-the-art performance on the PMC-Patients leaderboard, which suggests potential far-reaching applications of SciMult in the real world, such as matching patients with clinical trials for precision health.
>
> Finally, we need to argue that the contributions of a language model pre-training paper include not only its proposed new techniques but also its released model, which can be used by the public. Therefore, we feel the superior performance of a pre-trained language model on specific datasets is also an important criterion. As evidence, after BERT, many studies focus on developing more performant domain-specific backbone models – SciBERT [5], LEGAL-BERT [6], PhoBERT [7], and DarkBERT [8], to name just a few – were accepted by *ACL conferences. We fully support this and hope similar **empirical** contributions in our submission can be acknowledged as well.
>
> **References**
>
> [1] SciRepEval: A Multi-Format Benchmark for Scientific Document Representations. arXiv 2022.
>
> [2] Prefix-Tuning: Optimizing Continuous Prompts for Generation. ACL 2021.
>
> [3] Task-aware Retrieval with Instructions. Findings of ACL 2023.
>
> [4] Neighborhood Contrastive Learning for Scientific Document Representations with Citation Embeddings. EMNLP 2022.
>
> [5] SciBERT: A Pretrained Language Model for Scientific Text. EMNLP 2019.
>
> [6] LEGAL-BERT: "Preparing the Muppets for Court”. Findings of EMNLP 2020.
>
> [7] PhoBERT: Pre-trained language models for Vietnamese. Findings of EMNLP 2020.
>
> [8] DarkBERT: A Language Model for the Dark Side of the Internet. ACL 2023.

---

### Official Review · Reviewer_UpiD · 2023-08-04

**Soundness:** 3

**Excitement:**

3: Ambivalent: It has merits (e.g., it reports state-of-the-art results, the idea is nice), but there are key weaknesses (e.g., it describes incremental work), and it can significantly benefit from another round of revision. However, I won't object to accepting it if my co-reviewers champion it.

**Paper Topic And Main Contributions:**

The paper proposed a unified framework to solve all three kinds of scientific literature understanding tasks, which includes multi-label classification, link prediction and search. They adopted task-specific Transformer blocks for the unified model to overcome the limitations of backbone parameters-sharing. Specifically, two different types of Mixture-of-Experts are introduced for the Transformer architecture, which can route the input to different MHA and FFN sub-layers, respectively, when considering different tasks. Experimental results demonstrate the effectiveness of the proposed method.

**Reasons To Accept:**

1. The authors considered the problem of undesirable backbone parameters-sharing, avoiding suffering from the task interference.
2. The idea of using instruction tuning is reasonable for further enhancing the performance in the proposed basic unified-framework.
3. For each task, the paper tested on not only in-domain but also cross-domain evaluation datasets.
4. The paper is well written and easy to follow.

**Reasons To Reject:**

1. The assumption of the method is somewhat strong, that each label's name and definition must be both available.
2. The idea of multi-task learning and contrastive learning are not new, and there are no specific new insights for the task itself from the task nature perspective, which leads to a marginal contribution for the research field.

**Reproducibility:**

3: Could reproduce the results with some difficulty. The settings of parameters are underspecified or subjectively determined; the training/evaluation data are not widely available.

**Reviewer Confidence:**

4: Quite sure. I tried to check the important points carefully. It's unlikely, though conceivable, that I missed something that should affect my ratings.

---

> ### Author Rebuttal · Authors · 2023-08-29
>
> Thanks for your thoughtful review and feedback! We discuss the raised points as follows:
>
> 1. **Assumption on the availability of label names and definitions**
>
> This is a great question. First, we would like to clarify that **label definitions** are **not required** but optional during inference. In fact, in our experiments, the Geography and Psychology datasets do not have label definitions at all, thus classification on these two datasets relies on label names only. This is mentioned in the appendix (Lines 1003-1006, Page 12), and we will make this clearer in our revision.
>
> For model pre-training, because Zhang et al. [1] have released a large number of label definitions in CS and biomedicine domains, we leverage such information because we would like our pre-trained model to encode more semantic signals and to be more powerful. When applying our pre-trained model to downstream classification tasks, the model is flexible enough to take label names only. As shown in Table 3, on Geography and Psychology (where only label names are available), SciMult-MHAExpert outperforms all baselines consistently. On CS-Conference and Chemistry-MeSH, we conduct additional experiments by removing label definitions and let the model use label names only during inference. The results are as follows:
>
> | | CS-Conference | CS-Conference | CS-Conference | Chemistry-MeSH | Chemistry-MeSH | Chemistry-MeSH |
> | --- | --- | --- | --- | --- | --- | --- |
> | | **R@20** | **R@50** | **R@100** | **R@20** | **R@50** | **R@100** |
> | SPECTER 2.0 (Name Only) | 48.14 | 53.93 | 58.98 | 35.73 | 42.19 | 46.96 |
> | SPECTER 2.0 (Name+Definition) | 48.63 | 55.09 | 60.68 | 36.17 | 43.06 | 48.26 |
> | SciMult-MHAExpert (Name Only) | 52.57 | 61.27 | 67.84 | 37.83 | 46.47 | 52.59 |
> | SciMult-MHAExpert (Name+Definition) | 54.02 | 65.49 | 75.07 | 39.41 | 50.92 | 59.59 |
>
> We can observe that: after removing label definitions (the “Name Only” rows), although the performance of both SPECTER 2.0 and our SciMult-MHAExpert drops  (which is intuitive because we remove some useful information), SciMult-MHAExpert still consistently outperforms SPECTER 2.0, **even if SPECTER 2.0 uses label definitions**, supporting that our model is a performant general backbone model for scientific understanding.
>
> Second, the assumption of having **label names** in zero-shot text classification is widely accepted (e.g., [2, 3, 4]). After all, we need some information to characterize each label when annotated data is not available.
>
> 2. **Technical novelty of SciMult**
>
> We would like to clarify that the goal of this paper is not simply combining multi-task learning and contrastive learning. This is just a starting point of SciMult, fully covered by Section 3.1. In Sections 3.2 and 3.3, we make tremendous efforts to solve a general challenge in multi-task learning: task interference. Although the used techniques, including Mixture-of-Experts Transformers and instruction tuning, have been proposed in previous studies, they were not invented to mitigate task interference originally. In other words, we are using these techniques in a novel way to tackle a general problem. Then, in Section 3.4, we propose to sample hard negatives for each scientific literature understanding task. Note that we adopt different strategies for different tasks from the task nature perspective. In particular, we propose a new hard negative sampling strategy for paper classification based on citation/author/venue information, which is domain-specific and task-specific. This idea is not considered in previous studies as far as we know, and its effectiveness has been validated in Appendix F.1 and Table A5.
>
> Moreover, we believe a comprehensive benchmark study of scientific language models presented in this submission is also a contribution. Our study covers a wide range of tasks, compares various models in a controlled way, and provides insightful observations. In particular, our model achieves the new state-of-the-art performance on PMC-Patients, which is not included in the previous SciRepEval benchmark but has far-reaching applications in the real world, such as matching patients with clinical trials for precision health.
>
> **References**
>
> [1] Metadata-Induced Contrastive Learning for Zero-Shot Multi-Label Text Classification. WWW 2022.
>
> [2] Benchmarking Zero-shot Text Classification: Datasets, Evaluation and Entailment Approach. EMNLP 2019.
>
> [3] X-Class: Text Classification with Extremely Weak Supervision. NAACL 2021.
>
> [4] TaxoClass: Hierarchical Multi-Label Text Classification Using Only Class Names. NAACL 2021.

---

### Official Review · Reviewer_keSz · 2023-08-05

**Soundness:** 4

**Excitement:**

4: Strong: This paper deepens the understanding of some phenomenon or lowers the barriers to an existing research direction.

**Paper Topic And Main Contributions:**

The paper proposes SciMult, a multi-task contrastive learning framework, to jointly pretrain language models on datasets across multiple scientific literature understanding tasks (e.g., paper classification and literature search). The authors explore two different strategies, task-aware specialization, and instruction tuning, to mitigate interference between different tasks. The authors conduct comprehensive experiments on datasets across literature understanding tasks. Experimental results show that SciMult outperforms state-of-the-art language models and the effectiveness of task-aware specialization.

**Questions For The Authors:**

A. Given a specific dataset (task), does jointly pre-training on multiple datasets from different tasks improve upon training SciMult only on the one dataset (task)? For example, in Table 3, if we only train SciMult on CS-Conference, how does it perform on CS-Conference compared to SciMult-Vanilla which is jointly pre-trained on multiple tasks?

B. When training SciMult checkpoint, do you start from a random initialized Transformer or pre-trained ones such as SciBERT?

**Reasons To Accept:**

1. The paper conducted extensive experiments on a comprehensive collection of benchmarks. The experimental results show that the proposed method SciMult outperforms state-of-the-art scientific language models under in-domain settings and cross-domain settings.

2. The paper innovatively unifies different scientific literature understanding tasks into a contrastive learning framework.

3. The paper is clearly written and easy to follow.

**Reasons To Reject:**

1. According to Table 4 and Table A4, the proposed method underperforms or performs on par with SPECTER 2.0 on SciDocs (an in-domain dataset) when the evaluation setting aligns with SPECTER 2.0 (Linear and Reranking). This suggests that the performance improvement may be subject to the contrastive learning setting.

**Reproducibility:**

4: Could mostly reproduce the results, but there may be some variation because of sample variance or minor variations in their interpretation of the protocol or method.

**Reviewer Confidence:**

4: Quite sure. I tried to check the important points carefully. It's unlikely, though conceivable, that I missed something that should affect my ratings.

---

> ### Author Rebuttal · Authors · 2023-08-29
>
> Thanks for your thoughtful review and feedback! We discuss the raised points as follows:
>
> 1. **SciMult underperforming or performing on par with SPECTER 2.0 on SciDocs**
>
> First, we would like to mention the data leakage problem of SciDocs pointed out by Ostendorff et al. in [1]: “**40.5%** of SciDoc’s papers leaking into SPECTER’s training data”. This is also mentioned in our submission (Lines 1205-1210, Page 15). Because SPECTER, SciNCL, SPECTER 2.0, and SciMult all use SPECTER’s training data (i.e., the 819K citation prediction triplets) during pre-training, it becomes questionable how indicative the performance comparison can be if SciDocs is used for evaluation.
>
> Because of this, we exploit some other public datasets (i.e., PMC-Patients and Recommendation) to evaluate the link prediction task. Indeed, **none** of the texts in the Patient-to-Patient task and only **0.2%** of the papers in the Patient-to-Article task in PMC-Patients can be found in SPECTER’s training data. On these datasets, SciMult-MHAExpert consistently outperforms SPECTER 2.0 and other baselines, which is more persuasive in our view. Moreover, in fine-grained classification and search tasks where SciDocs is not used, SciMult-MHAExpert outperforms all baselines in almost all cases, with SciRepEval-Search as the only exception. This also implies the generalizability of our proposed model. We will make this clearer in our revision.
>
> 2. **Improvement upon training SciMult on only one dataset/task**
>
> Thank you for the great suggestion. To answer this question, we pre-train SciMult on CS-Journal only for the classification task. (We do use “CS-Conference” because it is one of our evaluation datasets, while “CS-Journal” is in our pre-training data.) We adopt the SciMult-Vanilla architecture, which is equivalent to SciMult-MHAExpert when there is only one task/dataset. The performance comparison on the classification task is shown as follows:
>
> | | CS-Conf | CS-Conf | CS-Conf | Chem-MeSH | Chem-MeSH | Chem-MeSH | Geo | Geo | Geo | Psych | Psych | Psych | Average |
> | --- | --- | --- | --- | --- | --- | --- | --- | --- | --- | --- | --- | --- | --- |
> | | **R@20** | **R@50** | **R@100** | **R@20** | **R@50** | **R@100** | **R@20** | **R@50** | **R@100** | **R@20** | **R@50** | **R@100** | |
> | SciMult-Vanilla (CS-Journal Only) | 54.44 | 64.11 | 72.29 | 31.99 | 35.50 | 38.67 | 53.59 | 55.76 | 58.11 | 45.44 | 47.36 | 49.42 | 50.56 |
> | SciMult-Vanilla (All Data) | 53.40 | 64.70 | 74.09 | 39.78 | 51.31 | 59.75 | 62.08 | 70.65 | 77.79 | 50.42 | 56.58 | 63.17 | 60.31 |
> | SciMult-MHAExpert (All Data) | 54.02 | 65.49 | 75.07 | 39.41 | 50.92 | 59.59 | 65.94 | 75.01 | 81.93 | 51.77 | 59.55 | 67.86 | 62.21 |
>
> We can observe that SciMult-Vanilla (CS-Journal Only) underperforms SciMult-Vanilla (All Data) and SciMult-MHAExpert (All Data) in all columns except R@20 on CS-Conference. This validates the motivation of multi-task learning that jointly training the model on multiple relevant tasks/datasets (even with the Vanilla architecture) can often outperform single-task learning. If we further devise architectures that can mitigate task interference (e.g., MHAExpert), the performance can be even higher. We will add this discussion in our revision.
>
> 3. **Initialization of SciMult**
>
> Sorry that we forgot to mention this in our submission. SciMult is initialized by the pre-trained PubMedBERT-abstract model (https://huggingface.co/microsoft/BiomedNLP-PubMedBERT-base-uncased-abstract). We will add this in our revision.
>
> **References**
>
> [1] Neighborhood Contrastive Learning for Scientific Document Representations with Citation Embeddings. EMNLP 2022.

---

### Meta-Review · Area_Chair_Nnh9 · 2023-09-19

**Recommendation:** 3

**Metareview:**

Summary:
The paper presents a unified framework for addressing three scientific literature understanding tasks: multi-label classification, link prediction, and search. To overcome the limitations of backbone parameter sharing, the authors employ task-specific Transformer blocks within the unified model. They introduce two types of Mixture-of-Experts in the Transformer architecture to route inputs to different Multi-Head Attention (MHA) and Feed-Forward Network (FFN) sub-layers based on the task at hand. Experimental results confirm the method's effectiveness.

Reason To Accept:
1.The paper is well-structured and presents its ideas in a clear and comprehensible manner.
2.The paper introduces an innovative approach by unifying various scientific literature understanding tasks within a contrastive learning framework. It devises four model variants to generate task-aware representations, facilitating shared knowledge acquisition across diverse tasks while mitigating interference from task-specific skills.
3.The authors address the issue of unwanted backbone parameters-sharing, preventing task interference, and propose a reasonable approach involving instruction tuning to further enhance the performance within the proposed unified framework.
4.The paper conducts extensive experiments across a wide range of benchmarks. The results demonstrate that the proposed SciMult method surpasses state-of-the-art scientific language models in both in-domain and cross-domain settings.

Reason To Reject:
1.According to Table 4 and Table A4, the proposed method underperforms or performs on par with SPECTER 2.0 on SciDocs (an in-domain dataset) when the evaluation setting aligns with SPECTER 2.0 (Linear and Reranking). This suggests that the performance improvement may be subject to the contrastive learning setting.
2.The assumption of the method is somewhat strong, that each label's name and definition must be both available.
3.The idea of multi-task learning and contrastive learning are not new, and there are no specific new insights for the task itself from the task nature perspective, which leads to a marginal contribution for the research field.

---

### Decision · Program_Chairs · 2023-10-07

**Decision:**

Accept-Findings

**Comment:**

Summary:
The paper presents a unified framework for addressing three scientific literature understanding tasks: multi-label classification, link prediction, and search. To overcome the limitations of backbone parameter sharing, the authors employ task-specific Transformer blocks within the unified model. They introduce two types of Mixture-of-Experts in the Transformer architecture to route inputs to different Multi-Head Attention (MHA) and Feed-Forward Network (FFN) sub-layers based on the task at hand. Experimental results confirm the method's effectiveness.

Reason To Accept:
1.The paper is well-structured and presents its ideas in a clear and comprehensible manner.
2.The paper introduces an innovative approach by unifying various scientific literature understanding tasks within a contrastive learning framework. It devises four model variants to generate task-aware representations, facilitating shared knowledge acquisition across diverse tasks while mitigating interference from task-specific skills.
3.The authors address the issue of unwanted backbone parameters-sharing, preventing task interference, and propose a reasonable approach involving instruction tuning to further enhance the performance within the proposed unified framework.
4.The paper conducts extensive experiments across a wide range of benchmarks. The results demonstrate that the proposed SciMult method surpasses state-of-the-art scientific language models in both in-domain and cross-domain settings.

Reason To Reject:
1.According to Table 4 and Table A4, the proposed method underperforms or performs on par with SPECTER 2.0 on SciDocs (an in-domain dataset) when the evaluation setting aligns with SPECTER 2.0 (Linear and Reranking). This suggests that the performance improvement may be subject to the contrastive learning setting.
2.The assumption of the method is somewhat strong, that each label's name and definition must be both available.
3.The idea of multi-task learning and contrastive learning are not new, and there are no specific new insights for the task itself from the task nature perspective, which leads to a marginal contribution for the research field.